# Questionnaire-Based Analysis of Adverse Events and Compliance with Malaria Chemoprophylaxis in Taiwan

**DOI:** 10.3390/jpm13020179

**Published:** 2023-01-19

**Authors:** Ching-Yun Lin, Ko Chang, Chai-Jan Chang

**Affiliations:** 1Department of Family Medicine, Kaohsiung Municipal SiaoGang Hospital, Kaohsiung 812, Taiwan; 2Department of Family Medicine, Kaohsiung Medical University Hospital, Kaohsiung 807, Taiwan; 3Tropical Medicine Center, Kaohsiung Medical University, Kaohsiung 807, Taiwan; 4Department of Internal Medicine, School of Medicine, College of Medicine, Kaohsiung Medical University, Kaohsiung 807, Taiwan; 5Department of Family Medicine, Kaohsiung Medical University, Kaohsiung 807, Taiwan

**Keywords:** malaria prophylaxis, compliance, side effects, mosquito, Taiwan

## Abstract

Malaria was eradicated in Taiwan in 1952; however, imported malaria cases are reported every year. The subtropical climate in Taiwan permits mosquito propagation and possible outbreaks of mosquito-borne diseases. The aim of this study was to investigate travelers’ compliance and side effects of malaria prophylaxis to prevent a malaria outbreak in Taiwan. In this prospective study, we enrolled travelers who visited our travel clinic before going to malarious areas. A total of 161 questionnaires were collected and analyzed. Associations between the occurrence of side effects and compliance with antimalarial drugs were analyzed. Adjusted odds ratios were calculated after adjusting for potential risk factors in multiple logistic regression analysis. Of the 161 enrolled travelers, 58 (36.0%) reported side effects. Insomnia, somnolence, irritability, nausea, and anorexia were associated with poor compliance. Mefloquine was not associated with more neuropsychological side effects than doxycycline. Multiple logistic regression analysis showed that chemoprophylaxis compliance was affected by a younger age, visiting friends and relatives, visiting the travel clinic more than 1 week before the trip, and preferring to use the same antimalarial regimen on the next trip. Our findings could provide information to travelers besides labeled side effects to improve compliance with malaria prophylaxis and consequently help to prevent malaria outbreaks in Taiwan.

## 1. Introduction

Malaria is an acute parasitic disease caused by the invasion of one or more of five species of genus Plasmodium through the bites from an infected female Anopheles mosquitoes, in which *Plasmodium falciparum* and *Plasmodium vivax* account for most cases. It remains one of the most deadly diseases worldwide: in 2021, nearly half of the world’s population was at risk of malaria, despite efforts to develop a vaccination, promote the usage of insecticide-treated nets, and integral preventive medical interventions [1]. The primary vector for malaria, Anopheles mosquitoes, are still found in 29 townships in Taiwan, as the hot and humid subtropic climate in Taiwan permits the mass propagation of mosquitoes. Taiwan was removed from the list of malaria-endemic countries by the World Health Organization (WHO) in 1965 [2]; however, around 30 imported cases of malaria are still recorded annually, and secondary transmission of malaria was reported as well [3,4]. An epidemiology report based on Taiwan CDC data between 2002 and 2013 revealed there were total 229 malaria cases; all of the cases were imported, 43% of these cases were African travel-related, and 44% of cases were Asian travel-related. The majority of mosquitos causing Plasmodium were *P. falciparum* (56%) [4]. Malaria is present in 20 countries in the southern, eastern, and southeastern regions of Asia as well as in the Asian Pacific area. The fact that, geographically, Taiwan is surrounded by malarious areas and the increasing number of tourists traveling to and from tropical Africa and South America, together with many immigrant workers and denizen brides from Southeast Asia, all potentially increase the risk of a malaria outbreak [5,6,7,8]. Highly drug-resistant malaria remains problematic in the Greater Mekong region where *P. vivax* and *P. falciparum* are the major parasites causing malaria [8]. The dormant features of *P. vivax* pose further potential of malaria spread in Taiwan. The island landscape protects Taiwan from the spread of infection to some degree, but not totally, as seen with the COVID-19 epidemic. In addition, rabies was eradicated from Taiwan in 1961 but re-emerged in 2013 [9,10]. Therefore, active preventive measures against imported malaria by international travelers to endemic regions cannot be delayed.

According to the latest world malaria report, there were 247 million cases of malaria in 2021, with estimated global deaths of 619,000. Mostly, cases are concentrated in Africa. Reported cases and deaths are increased compared to an estimated 214 million infections and 438,000 deaths in 2015 [11]. Nonimmune travelers, travelers visiting friends and relatives, and young children are susceptible to contracted malaria, especially that causing severe illness, deaths, and even local spread of malaria. Imported malaria should be monitored carefully by national authorities. According to the report regarding US travelers from 2004 to 2014, imported malaria infections numbered 17,471, and *P. falciparum* or *P. vivax* comprised the majority of infections [12]. One European report supposed that under-reporting of imported malaria was probably common; according to a 2010 WHO report, there were 6244 cases of malaria imported to Europe, but the true number might be six times higher [13,14]. Today, malaria chemoprophylaxis maintains its effectiveness against malaria infection with the right dosage and the right antimalarial drugs, considering high drug resistance. That is efficient to prevent imported malaria by travelers as well.

We conducted this study at the travel clinic in Kaohsiung Municipal Siaogang Hospital, which is directed by the Taiwan Centers for Disease Control (CDC). The clinic provides pretrip counseling and risk evaluation, prophylactic medications, and vaccinations. Services including medications, vaccinations, and counseling, and fees are all self-paid and not covered by the National Health Insurance program of Taiwan. Few studies have investigated compliance with malaria chemoprophylaxis and the side effects of antimalarial agents in Taiwan compared to other countries [15,16,17,18,19]. Therefore, the aim of this study was to investigate these issues in travelers who visited our clinic before going to malarious areas. In Taiwan, we have three options, including the use of mefloquine, doxycycline, and atovaquone-proguanil for malaria prophylaxis [20,21,22]. We aim to know the characteristic and compliance of antimalarial medication of our travelers. We also aim to know the detailed side effects profile and occurrence rates for Taiwanese reference.

## 2. Materials and Methods

### 2.1. Study Design and Participants

In the prospective study, we enrolled travelers who visited the travel clinic between January 2017 and December 2019 for malaria chemoprophylaxis. According to CDC recommendations, travelers could choose one from mefloquine, atovaquone-proguanil, doxycycline, chloroquine, primaquine, and tafenoquine for primary malaria prophylaxis [23]. In Taiwan, we have mefloquine, a once weekly regimen, and 2 kinds of a once daily regimen to choose from: doxycycline and atovaquone-proguanil. We interviewed 173 travelers, of whom 7 on atovaquone-proguanil were excluded because of the small sample size, and 5 were excluded because they were lost to follow-up after their trip.

All enrolled 161 travelers were Taiwanese, and we communicated verbally in Chinese; the questionnaire was also written in Chinese. Informed consent was obtained from every participant in the study during the first clinic visit. This study was approved by Kaohsiung Medical University Hospital Institution Review Board, KMUH-IRB-970496.

### 2.2. Study Questionnaire

The travelers were asked to complete questionnaires at the clinic during their first visit, which included questions on demographic profiles, details of the trip, and knowledge, attitudes, and practices of malaria. Pretrip education and discussion were performed with the same team consisting of two doctors, and pre- and posttrip data were collected by the same assistant. All costs for medications and doctors’ visits were based on standard travel clinic charges without any reduction. We contacted 161 returning travelers via phone calls and collected information about adherence and the occurrence of side effects of antimalarial agents.

Adherence to the chemoprophylaxis regimen was recorded according to self-reported use. In the questionnaire, adherence to antimalarial agents was assessed as (1) completed treatment, (2) skipped doses sometimes, and (3) early termination. We defined compliant travelers as those who completed treatment without skipping any doses or terminating treatment early.

We recorded side effects as reported by the travelers and classified them as neuropsychological (dizziness, nightmares, headache, anxiety, insomnia, somnolence, irritability, and depression) and non-neuropsychological (nausea, anorexia, abdominal pain, palpitations, and reflux esophagitis) side effects. The travelers were asked to grade the severity as (1) mild, feeling different; (2) moderate, interfering with daily activities sometimes; and (3) severe, interrupting daily activities.

### 2.3. Statistical Analysis

Continuous and categorical data are presented as mean (standard deviation) and frequency (percentage), respectively, and they were compared using the t test and chi-square test or Fisher’s exact test to identify associations between the occurrence of side effects and compliance with the antimalarial agents. Adjusted odds ratios (ORs) were calculated after adjusting for potential risk factors using multiple logistic regression analysis. All statistical analyses were performed using SAS version 9.4 (SAS Institute, Cary, NC, USA). A *p* value < 0.05 was considered to be statistically significant.

## 3. Results

### 3.1. Sociodemographic Characteristics and Details of the Trip

Of the 161 enrolled travelers (54.7% male, 45.3% female), 31.1% were younger than 30 years, 29.8% were aged 30–45 years, 31.7% were aged 45–60 years, and 7.5% were older than 60 years. The most frequently visited destination was Africa (46%), followed by Oceania islands (19.9%). In addition, 32.9% of the travelers stayed in malarious areas for less than 1 week, most travelers (53.4%) stayed for 1 week to 4 weeks, and only 3.7% stayed for more than 4 weeks. Regarding education level, 55.9% of the travelers had a bachelor’s degree, and 19.3% had a master’s degree and/or Ph.D., which is relatively high compared to the general population in Taiwan. Concerning the purpose of the visit, most travelers went to malarious areas for work (39.8%), followed by vacation (31.7%). The working status of the travelers was evenly distributed, including businesspeople (19.3%), teachers (16.1%), others (14.2%), students (13.7%), civil servants (11.2%), unemployed individuals (9.3%), and engineers (8.1%). Overall, 21.7% of the travelers had traveled to malarious regions, and 21.7% had taken antimalarial medications accordingly before our travel clinic visit. More travelers chose mefloquine (72.7%) than doxycycline (27.3%) as their antimalarial medication. We divided the travelers into three groups (<1 week, 1~4 weeks, >4 weeks) according to how long before departure they planned the trip, as well as when they visited the travel clinic, to analyze the best time to visit the travel clinic. Most of the travelers (77%) planned the trip more than 4 weeks in advance. In addition, 46.6% of the travelers visited our clinic 1~4 weeks before the trip, 32.9% less than 1 week before the trip, and 19.3% more than 4 weeks before the trip. With regards to a travel companion, most travelers (46.6%) traveled alone. The chemoprophylaxis compliance rate is 59.6% in total: 62.4% in the mefloquine group, and 52.3% in the doxycycline group, accordingly (Table 1).

### 3.2. Side Effects between Mefloquine and Doxycycline

We recorded side effects from the antimalarial agents as reported by the travelers (Table 2). Concerning the frequency of reported side effects, 58 of the 161 travelers (36%) reported at least one side effect. The most commonly reported side effect was nausea (17.4%), followed by dizziness (12.4%).

With regards to neuropsychological side effects, 29 of 119 (24.8%) travelers reported neuropsychological side effects after taking mefloquine, compared to 12 of 44 (27.3%) who took doxycycline, and the difference was not statistically significant (*p* = 0.7469) (Table 2). In addition, there was no significant difference in the occurrence rate of all side effects between those who took mefloquine and those who took doxycycline (33.3% vs. 43.2%, *p* = 0.2460). Mefloquine was not associated with a higher risk of developing neuropsychological side effects than doxycycline. Travelers on doxycycline reported dizziness and anorexia more, and anorexia might impede compliance (*p* = 0.0051).

### 3.3. Relationship between Drug Compliance and Side Effects

With regards to the relationship between drug compliance and the occurrence of side effects, insomnia, somnolence, irritability, nausea, and anorexia were associated with poor compliance in univariate analysis (Table 3). None of these side effects remain significant to influence drug compliance in the result of multiple logistic regression.

### 3.4. Chemoprophylaxis Compliance

With regards to chemoprophylaxis compliance, 59.6% of the 161 travelers were compliant, and 40.4% were not. Most of the noncompliant travelers did not specify the reason (46.2%), followed by forgetfulness (30.8%). In addition, 81.4% of the travelers preferred a once weekly regimen, and the preference of antimalarial drugs did not affect compliance. However, after taking their antimalarial agents, 20.5% (9/44) of the travelers on doxycycline stated that they would prefer to use a different regimen next time, compared to only 6.8% (8/117) of those who used mefloquine. Good compliance (95.8%) was found in the travelers who would choose the same antimalarial medication for future trips.

### 3.5. Effect of Demographic Characteristics on Compliance

In our study, drug compliance was affected by the age of travelers, travel destination, purpose of travel, time interval between travel clinic visiting and departure, and use of same antimalaria drug next trips with univariate analysis. Compared to travelers aged 30–45 years old, age younger than 30 years old and older than 60 years old were less compliant. Travelers were more compliant while traveling to Africa compared to other regions. Travelers for vacation and business had fair compliance, and travelers for visiting relatives and friends had worse compliance. In the results of multiple logistic regression analysis, they complied better with chemoprohylaxis while traveling for vacation. The differences of adherence were not significant in the length of stay, the education of travelers, the traveling companions, previous experiences of both traveling to malarious areas and taking antimalarial drugs, and when their trips planned. No difference in compliance was observed between mefloquine and doxycycline in our study. Regarding the impact of demographic characteristics on compliance, significant influential factors included age, purpose of visit, travel destination, when to visit the travel clinic, and the choice of regimen for the next trip (Table 4). These variables were included in the multiple logistic regression analysis, which showed that the independent factors for worse compliance were younger age, vacation as the travel purpose, visiting the travel clinic more than 1 week before their trip, and favoring a different antimalarial agent for the next trip. In addition, the adjusted OR for visiting the travel clinic more than 1 month before the trip was 0.17 (*p* = 0.01), compared to 0.33 (*p* = 0.0495) for visiting the travel clinic 1 week–1 month before the trip, suggesting that better compliance was related to visiting the travel clinic within 1 week of departure.

## 4. Discussion

The main purpose of this study was to investigate compliance and side effects of antimalarial agents in Taiwanese travelers visiting other countries. Our results showed that the younger travelers tended to be more noncompliant. Previous studies have reported that younger age and extensive travel experience are risk factors for noncompliance [24,25,26]. However, a previous travel history to malarious areas did not affect compliance in our study. It is possible most experienced travelers did not visit our travel clinic and thus were not included in our study. 

The purpose of the trip and visited regions had different influences on compliance in this study. Visiting friends and relatives (VFR) was associated with noncompliance in our study, whereas taking a vacation was associated with compliance (adjusted OR 3.86, *p* = 0.0388). These findings are similar to studies conducted in airports in Spain [27], the USA [28], Zimbabwe [29] (malaria-endemic area), and Korea [30] in which vacation as the purpose of the trip was associated with fair compliance, while VFR was associated with poorer compliance. VFR travelers have also been reported to have high prevalence rates of febrile illness (malaria) and low rates of seeking pretravel advice [31]. Therefore, efforts should be made to promote the services of travel clinics to travelers going on vacation, since they tend to care about and follow instructions. In addition, travelers VFR should be educated with regards to the severity of a delayed diagnosis of malaria and prevalence of multidrug-resistant malaria. This could be challenging, as the compliance rate was only 1/7 (14.3%) in the travelers VFR in this study.

The most frequently visited malarious areas of Taiwanese travelers differ from those of Spanish (84.6%) [27] and American (72%) [28] travelers, who usually visit low-malaria-endemic regions such as South America and Asia more. Most Taiwanese travelers in this study (74/161, 46.0%) traveled to Africa, a high-malaria-endemic region, and most showed a higher tendency to be compliant (52/74, 70.3%). This finding suggests that the perceived risk of a destination is an important factor for compliance.

The risk of contracting malaria is higher with longer duration of travel; however, studies of systemic review showed inconsistent results about compliance with duration of stay in malarious areas. Inconsistency is possibly due to the lack of unified definitions and categories in every studies. Longer stay was reported in numerous studies as a significant influential factor of poorer compliance [32]. Take one Netherlands study for example: travelers were significantly more compliant with shorter stays (mean 20 days) than longer stays (mean 32 days) [33]. In our study, the relationship of the length of stay (<7 days, 7–28 days, and >28 days) and drug compliance was not significant in univariate analysis. It could be explained that Taiwanese travelers tend to follow regimen strictly regardless of the length of stay.

We also found that the timing of visiting the travel clinic influenced compliance. Our results showed that those who visited the clinic less than 1 week before the trip had the best compliance, and that a longer duration was associated with worse compliance. We should alert all travelers by phone, messaging, and email to remind them frequently upon their departure.

Many reports have reported an association between mefloquine and a higher incidence of neuropsychological side effects [18,19,34,35]. However, we did not find that mefloquine was associated with more neuropsychological side effects than doxycycline. This may be explained by the low dose used in prophylaxis, because most studies suggest that neuropsychologic toxicity from mefloquine is dose-dependent. Toovey et al. [34] suggested a number of explanatory mechanisms for the psychoactive and neurotoxic effects of mefloquine and that female users, lower body mass, and the tendency for the drug to accumulate in the central nervous system were associated with an increased risk. Some studies have reported weak correlations between mefloquine and neuropsychological side effects. A Thai pooled analysis of the side effects of mefloquine for the treatment of malaria on the Thai–Myanmar/Cambodia borders in 19,850 individuals indicated that serious neuropsychiatric side effects were rare [36]. In addition, a Japanese study of 1876 self-defense members who received mefloquine for 6-month prophylaxis in 2002 reported that mefloquine was generally safe unless an individual was predisposed to neuropsychiatric illness [37]. Another possible explanation may be racial differences, as we also observed similar outcomes to the Japanese and Thai studies. However, due to the observational characteristics of our study, it should be addressed more carefully.

In our study, malaria chemoprophylaxis compliance was affected by side effects from the antimalarial drugs including insomnia, somnolence, irritability, nausea, and anorexia. However, in multiple logistic regression analysis, the effect was not significant. Previous studies have reported that poor compliance is related to the occurrence of side effects from antimalarial agents [24,38,39,40]. Thus, travel clinics may prepare medicine for relieving symptoms, give loading doses of the antimalarial agents before travel, and change to an alternative antimalarial drug in cases of intolerable side effects.

The travelers who expressed a preference to receive a different antimalarial agent on the next trip were noncompliant in this study (adjusted OR 0.07, *p* = 0.001). Hence, an unsatisfactory experience with antimalaria drugs appears to be a reason for poor compliance. A review study concluded that compliance was better with weekly prophylaxis than daily prophylaxis overall; however, the difference was not significant between mefloquine (71.97%) and doxycycline (71.9%) [41]. It corresponds to our result of 62.4% compliance with mefloquine and 52.3% with doxycycline. Some studies reported that compliance in military troops was worse than civilians [32], as seen in studies of Marie-Aude Creach et al. in which the prophylaxis compliance in troops was 56.6% [42]. In contrast, there was a different result seen in a Sri Lankan study with a compliance rate 90.3% in the civilian group and 100% in the military group [43]. In our study, the compliance rate (59.6%) is much lower than civilian compliance rates seen in previous studies. That we can see chemoprophylaxis compliance in Taiwanese civilian travelers is not good enough; we have to work harder to enhance civil travelers’ adherence to prophylaxis. The appropriate choice of drugs for chemoprophylaxis is affected by several issues, including drug efficacy, tolerance, convenience, and cost [44,45], so clinicians should provide travelers with appropriate information and recommendations. A better understanding of travelers’ knowledge, attitudes, and practices of malaria and perceived risk is imperative for travel clinic practice.

Compared to previous studies from outbound military troops and airports, our study was focus on travelers visiting the travel clinic with the same setting and same group of doctors. There might be less data collecting and processing bias. To examine the evidence better, more travelers should be enrolled in the future.

Recent studies found to be important the personal socio-psychologic factors [21,32,33,41,42] that could be particularly influenced by different races and countries, which is why we have to know Taiwanese’s determinant factors of chemoprophylaxis compliance.

## 5. Conclusions

The result of our study suggests that mefloquine is not associated with more neuropsychological side effects in Taiwanese, so both physicians and travelers might use mefloquine, once weekly dosage, with less worry. Based on the identified risk factors of noncompliance, we could develop a comprehensive approach with the implementation of prophylactic medication directing side effects of chemoprophylaxis, reminders within one week before departure, and a reinforcement program toward a younger age group and VFR (visiting friends and relatives) travelers. With ongoing follow-up of more travelers, we hope to give our travelers optimal antimalaria recommendations according to global guidelines and our domestic research results. With this information, real-time reminders and medical responses will be our goal. The limitations in this study include the small number of study subjects, the short follow-up time, and that they were all recruited from a travel clinic rather than all travelers at risk. Our study is observational rather than randomized controlled trials, meaning we should explain the result carefully and follow evidence-based suggestions.

## Figures and Tables

**Table 1 jpm-13-00179-t001:** Comparisons of sociodemographic characteristics and details of the trip between the compliant and noncompliant groups. *N* = 161.

	All Travelers	Compliant (1)	Noncompliant (2 + 3 + 4)	*p* Value ^1^
*N*	161 ^2^	96	65	
**Age** **, *N* (%)**				
<30	50 (31.1)	16 (16.7)	34 (52.3)	
30–45	48 (29.8)	40 (41.7)	8 (12.3)	
45–60	51 (31.7)	34 (35.4)	17 (26.2)	
>60	12 (7.5)	6 (6.3)	6 (9.2)	**<0.0001**
Gender, n (%)				
Male	88 (54.7)	54 (56.3)	34 (52.3)	
Female	73 (45.3)	42 (43.8)	31 (47.7)	0.6220
**Travel destination** **, *N* (%)**				
Middle and South America	14 (8.7)	3 (3.1)	11 (16.9)	
Southeast Asia	16 (9.9)	9 (9.4)	7 (10.8)	
Africa	74 (46.0)	52 (54.2)	22 (33.8)	
India	25 (15.5)	17 (17.7)	8 (12.3)	
Oceania islands ^3^	32 (19.9)	15 (15.6)	17 (26.2)	**0.0048**
Length of stay, n (%)				
<7 days	53 (32.9)	37 (38.5)	16 (24.6)	
7–28 days	86 (53.4)	45 (46.9)	41 (63.1)	
>28 days	22 (13.7)	14 (14.6)	8 (12.3)	0.1145
**Purpose of visit** **, *N* (%)**				
Studies	1 (0.6)	0 (0.0)	1 (1.5)	
Business/work	64 (39.8)	45 (46.9)	19 (29.2)	
Vacation	51 (31.7)	32 (33.3)	19 (29.2)	
Visiting friends/relatives	7 (4.3)	1 (1.0)	6 (9.2)	
Volunteer work	26 (16.1)	11 (11.5)	15 (23.1)	
Religious visits	2 (1.2)	2 (2.1)	0 (0.0)	
Others	10 (6.2)	5 (5.2)	5 (7.7)	**0.0156**
Education, *N* (%)				
Junior high school	7 (4.3)	4 (4.2)	3 (4.6)	
Senior high school	9 (5.6)	7 (7.3)	2 (3.1)	
College	24 (14.9)	17 (17.7)	7 (10.8)	
University	90 (55.9)	47 (49)	43 (66.2)	
Master or/and Ph.D.	31 (19.3)	21 (21.9)	10 (15.4)	0.2487
Have you ever traveled in malarial regions before?. *N* (%)				
Yes	35 (21.7)	24 (25.0)	11 (16.9)	
No	126 (78.3)	72 (75.0)	54 (83.1)	0.2228
Have you ever taken antimalarial drugs before?. *N* (%)				
Yes	35 (21.7)	20 (20.8)	15 (23.1)	
No	126 (78.3)	76 (79.2)	50 (76.9)	0.7349
Antimalarial drugs used during this trip, *N* (%)				
Mefloquine	117 (72.7)	73 (76.0)	44 (67.7)	
Doxycycline	44 (27.3)	23 (24.0)	21 (32.3)	0.2435
How many days before departure did you plan for this trip?. *N* (%)				
1–7 days	11 (6.8)	8 (8.3)	3 (4.6)	
7–28 days	26 (16.1)	17 (17.7)	9 (13.8)	
>28 days	124 (77.0)	71 (74)	53 (81.5)	0.4889
**How many days before departure did you visit travel clinic?, *N* (%)**				
1–7 days	53 (32.9)	37 (38.5)	16 (24.6)	
7–28 days	77 (47.8)	45 (46.9)	32 (49.3)	
>28 days	31 (19.3)	14 (14.6)	17 (26.2)	**0.0473**
Whom did you accompany during this trip?, *N* (%)				
Spouse	19 (11.8)	6 (6.3)	13 (20.0)	
Friends	34 (21.1)	23 (24.0)	11 (16.9)	
Relatives	16 (9.9)	8 (8.3)	8 (12.3)	
Colleagues	15 (9.3)	7 (7.3)	8 (12.3)	
Alone	75 (46.6)	50 (52.1)	25 (38.5)	
Spouse + Friends	1 (0.6)	1 (1.0)	0 (0.0)	
Spouse + Relatives	1 (0.6)	1 (1.0)	0 (0.0)	0.0695
Preferred regimen, *N* (%)				
Once daily (1)	30 (18.6)	17 (17.7)	13 (20.0)	
Once weekly (2)	131 (81.4)	79 (82.3)	52 (80.0)	0.7141
**Will you take the same antimalarial agent next trip?*N* (%)**				
**Yes (1)**	138 (85.7)	92 (95.8)	46 (70.8)	
**No, try other medicine (2)**	17 (10.6)	4 (4.2)	13 (20.0)	
**No use of any medicine (3)**	6 (3.7)	0 (0.0)	6 (9.2)	**<0.0001**
Side effects (AF-AR)				
Score = 0	103 (64.0)	66 (68.8)	37 (56.9)	
Score > 0	58 (36.0)	30 (31.3)	28 (43.1)	0.1251
Neuropsychological side effects				
Score-Neu = 0	120 (74.5)	74 (77.1)	46 (70.8)	
Score-Neu > 0	41 (25.5)	22 (22.9)	19 (29.2)	0.3669
Non-neuropsychological side effects				
Score-Non-Neu = 0	126 (78.3)	80 (83.3)	46 (70.8)	
Score-Non-Neu > 0	35 (21.7)	16 (16.7)	19 (29.2)	0.0579

^1^*p* values, chi-square test; Categories are specified in bold if *p* < 0.05. ^2^ There were 161 effective questionnaires in total; ^3^ The large number of visitors to Oceania islands was due to cooperation with volunteer services in the Solomon Islands provided by our hospital.

**Table 2 jpm-13-00179-t002:** Associations between side effects and antimalarial drugs. *N* = 161.

Side Effect ^2^			Mefloquine	Doxycycline	
	Mild (1)	Moderate (2)	*N* = 117	*N* = 44	*p* Value ^1^
Dizziness	18 (11.2)	2 (1.2)	12 (10.3)	8 (18.2)	0.1870
Nightmare	5 (3.1)	0 (0.0)	4 (3.4)	1 (2.3)	1.0000
Headache	15 (9.3)	0 (0.0)	10 (8.5)	5 (11.4)	0.5561
Anxiety	13 (8.1)	1 (0.6)	11 (9.4)	3 (6.8)	0.7598
Insomnia	10 (6.2)	0 (0.0)	9 (7.7)	1 (2.3)	0.2879
Somnolence	4 (2.5)	2 (1.2)	3 (2.6)	3 (6.8)	0.3465
Irritability	8 (5.0)	0 (0.0)	7 (6.0)	1 (2.3)	0.4482
Depression	3 (1.9)	0 (0.0)	2(1.7)	1 (2.3)	1.0000
Nausea	25 (15.5)	3 (1.9)	18 (15.4)	10 (22.7)	0.3502
**Anorexia**	**9 (5.6)**	**2 (1.2)**	**4 (3.4)**	**7 (15.9)**	**0.0051**
Abd. Pain	9 (5.6)	0 (0.0)	7 (6.0)	2 (4.5)	1.0000
Palpitation	6 (3.7)	1 (0.6)	5 (4.3)	2 (4.5)	1.0000
Others	5 (3.1)	0 (0.0)	3 (2.6)	2 (4.5)	0.6147
Side effects (total)					
* Score = 0			78 (66.7)	25 (56.8)	
Score > 0			39 (33.3)	19 (43.2)	0.2721
Neuropsychological side effects ^3^					
Score = 0			88 (75.2)	32 (72.7)	
Score > 0			29 (24.8)	12 (27.3)	0.8395
Non-neuropsychological side effects ^4^					
Score = 0			94 (80.3)	32 (72.7)	
Score > 0			23 (19.7)	12 (27.3)	0.2934

^1^*p* value: Fisher’s exact test; Categories are specified in bold if *p* < 0.05. ^2^ The side effects experienced are multichoice answers; ^3^ Neuropsychological side effects such as dizziness, nightmare, headache, anxiety, insomnia, somnolence, irritability, depression; ^4^ Non-neuropsychological side effects such as nausea, anorexia, abdominal pain, palpitation, and others side effects. * The score is calculated by total amount of the experienced side effects. No severe side effect was observed.

**Table 3 jpm-13-00179-t003:** Associations between side effects and drug compliance. *N* = 161.

Side Effects ^1^	Compliant	Noncompliant	Chi-Square Test	Fisher’s Exact Test
	*N* = 96	*N* = 65	*p* Value	*p* Value
Dizziness	10 (10.4)	10 (15.4)	0.3484	0.4657
Nightmare	4 (4.2)	1 (1.5)	0.3456	0.6490
Headache	9 (9.4)	6 (9.2)	0.9754	1.0000
Anxiety	7 (7.3)	7 (10.8)	0.4423	0.5703
**Insomnia**	**2 (2.1)**	**8 (12.3)**	**0.0084**	**0.0155**
**Somnolence**	**1 (1.0)**	**5 (7.7)**	**0.0288**	**0.0398**
**Irritability**	**1 (1.0)**	**7 (10.8)**	**0.0053**	**0.0077**
Depression	0 (0.0)	3 (4.6)	0.0336	0.0640
**Nausea**	**11 (11.5)**	**17 (26.2)**	**0.0158**	**0.0199**
**Anorexia**	**2 (2.1)**	**9 (13.8)**	**0.0037**	**0.0076**
Abd. Pain	3 (3.1)	6 (9.2)	0.0980	0.1593
Palpitation	2 (2.1)	5 (7.7)	0.0868	0.1194
Others	4 (4.2)	1 (1.5)	0.3456	0.6490

^1^ The side effects experienced were multichoice answers. Side effects is specified in bold in *p* < 0.05.

**Table 4 jpm-13-00179-t004:** Results of multiple logistic regression analysis. *N* = 161.

	Compliant (1)	Noncompliant (2 + 3 + 4)	*p* Value	OR (95% CI)	*p* Value	Adjusted OR (95% CI)	*p* Value
*N*	96	65					
**Age, *N*(%)**							
<30	16 (16.7)	34 (52.3)		0.27 (0.12–0.59)	0.0011	0.34 (0.13–0.90)	**0.0294**
30–45	40 (41.7)	8 (12.3)		2.87 (1.15–7.19)	0.0239	1.99 (0.65–6.08)	0.2269
>45	40 (41.7)	23 (35.4)	<0.0001	1.00		1.00	
Travel destination, *N* (%)							
Non-Africa regions	44 (45.8)	43 (66.2)		1.00		1.00	
Africa	52 (54.2)	22 (33.8)	0.0111	2.31 (1.20–4.43)	0.0119	1.51 (0.63–3.60)	0.3572
**Purpose of visit, *N* (%)**							
others ^1^	8 (8.3)	12 (18.5)		1.00		1.00	
Business/work	45 (46.9)	19 (29.2)		3.55 (1.25–10.08)	0.0172	3.75 (0.99–14.12)	0.0508
Vacation	32 (33.3)	19 (29.2)		2.53 (0.88–7.29)	0.0865	3.86 (1.07–13.91)	**0.0388**
Volunteer work	11 (11.5)	15 (23.1)	0.0215	1.10 (0.34–3.60)	0.8748	4.07 (0.95–17.51)	0.0591
**How many days before departure did you visit travel clinic?, *N* (%)**							
1–7 days	37 (38.5)	16 (24.6)		1.00		1.00	
7–28 days	45 (46.9)	30 (46.2)		0.65 (0.31–1.37)	0.2558	0.33 (0.11–0.99)	**0.0495**
>28 days	14 (14.6)	17 (26.2)		0.36 (0.14–0.89)	0.0276	0.17 (0.04–0.65)	**0.0100**
Unknown	0 (0.0)	2 (3.1)	0.0473	-	-	-	-
Whom did you accompany during this trip?, *N* (%)							
More accompany	46 (47.9)	40 (61.5)		1.00		1.00	
Alone	50 (52.1)	25 (38.5)	0.0891	1.74 (0.92–3.3)	0.0903	1.08 (0.41–2.86)	0.8721
**Will you take the same antimalarial drug next trip?, *N* (%)**							
Yes (1)	92 (95.8)	46 (70.8)		1.00		1.00	
No, try other medicine (2)	4 (4.2)	13 (20.0)		0.15 (0.05–0.50)	0.0018	0.09 (0.02–0.44)	**0.0026**
No use of any medicine (3)	0 (0.0)	6 (9.2)	<0.0001	-	-	-	-

OR: odds ratio; CI: confidence interval; Adjusted ORs was calculated using a multiple logistic regression model; OR > 1, *p* < 0.05, Compliant; OR < 1, *p* < 0.05, Noncompliant; ^1^ Purpose of visit including study, visiting relatives/friends, and religious reasons were categorized into “others” due to the small numbers. Categories are specified in bold if *p* < 0.05.

## Data Availability

Not applicable.

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
