# Peer review of "Questionnaire-Based Analysis of Adverse Events and Compliance with Malaria Chemoprophylaxis in Taiwan"

_jpm, 2023, doi:10.3390/jpm13020179_

Round 1

Reviewer 1 Report

The manuscript entitled "Questionnaire-based Analysis of Adverse Events and Compliance with Malaria Chemoprophylaxis in Taiwan". Title, abstract and overall rationale of work to some extent is good. However, there are still some major concerns, which needs to be addressed and needs substantial revision.

1) Author must be include two more keywords such as mosquito and Taiwan

2) Introduction section: This section is too short and author need to elaborate. Author must be incorporate world malaria current data about mortality rate see this article (DOI: 10.1007/s13205-021-03022-0) or Moreover, author also need to explain all these drugs side effect in the introduction section and mention other antimalarial drugs usages in Taiwan. I suggest author incorporate which kind of Plasmodium species found mostly in traveler for example, P. falciparum or P. vivax.

3) Results section: Line number 108-109 author must be write other antimalrial drugs name.

I read all results and I found author describe about the side effect of all these drugs unfortunately, all these are well known and they already mention about their side effect of these antimalarial drugs in CDC/WHO malaria website. My question is why author choose this kind of study and what is benefit of the scientific community kindly explain.

4) Discussion section: Author need expend discussion and much more explanations and interpretations must be added for the results, which are not enough at all. It is suggested to compare the results of the present research with some similar studies which is done before.

5) Why author did not write anything about conclusion. Author must be write conclusion of this study and also need to write scientific important of this study.

6) There are some of punctuation and typographical errors throughout in the manuscript. Please correct it

7) References is not arrange and please correct it.

Author Response

I would like to express my sincere gratitude to the two reviewers for their professional analysis and constructive suggestions to make my paper more readable, and for the opportunity to revise. The reviewers’ suggestions and my revisions are outlined as follows. All emendations are marked in highlighted words in the text.  

Outline of Reviewers’ suggestions and the Author’s Correction

Reviewer #1

  • Author must be include two more keywords such as mosquito and Taiwan

  Keywords Taiwan and mosquito are added.

  • Introduction section: This section is too short and author need to elaborate. Author must be incorporate world malaria current data about mortality rate see this article (DOI: 10.1007/s13205- 021-03022-0) or Moreover, author also need to explain all these drugs side effect in the introduction section and mention other antimalarial drugs usages in Taiwan. I suggest author incorporate which kind of Plasmodium species found mostly in traveler for example, P. falciparum or P. vivax.

   It is included the information in discussion section; the added/corrected part are in red color.

  • Results section: Line number 108-109 author must be write other antimalrial drugs name. I read all results and I found author describe about the side effect of all these drugs unfortunately, all these are well known and they already mention about their side effect of these antimalarial drugs in CDC/WHO malaria website. My question is why author choose this kind of study and what is benefit of the scientific community kindly explain. 2022/12/15

   We also have travelers use malarone, however, because of small numbers, they are excluded from this study.

   Previous studies report the relationship of worse CP compliance and side effects. We want to know side effects profile of Taiwanese and how these side effects affect Taiwanese CP compliance.  

  • Discussion section: Author need expend discussion and much more explanations and interpretations must be added for the results, which are not enough at all. It is suggested to compare the results of the present research with some similar studies which is done before.

  More of previous researches are included and compared in discussion parts.  

  • Why author did not write anything about conclusion. Author must be write conclusion of this study and also need to write scientific important of this study.

  Conclusion is added to include the importance of this study.  

  • There are some of punctuation and typographical errors throughout in the manuscript. Please correct it

  I have checked thoroughly and corrected many written errors.

  • References is not arrange and please correct it.

  References are rearranged.

Reviewer 2 Report

Overall, this article is very well written and may be of interest to physicians prescribing chemoprophylaxis for travelers in malaria-endemic areas. Minor corrections need to be made to the article as well as a better discussion of the limitations of the study and adding a conclusion.

Multiples citations should follow usual rules. For example [17][18][19] or [30,31,32] should be cited as [17-19] or [30-32]

Abstract : Line 18-19 : « Of the 161 enrolled travelers, 103 (64%) reported side effects ». This appears to be in contradiction with the results (line 124) « Concerning the frequency of reported side effects, 66 of the 161 travelers (40.7%) reported at least one side effect ». Can you explain the difference or correct the sentence?

Line 107-108: « Overall, 32.7% of the travelers traveled to malarious regions, and 32.7% took antimalarial medications accordingly. » Could you please specify 32.7% of which population?

Line 116-117 : Table 1 title : specify n=161

Table 1: Replace doxycycline by Doxycycline; female by Female

Side effects :

-       If so, specify that no severe side effects were observed or describe it.

-       Define in methods how was calculated the score. In the table, the total amount of travelers with score>0 is 58 and you say in the text that 66 of the 161 travelers reported side effects : What can explain the difference ?

-       P values: when n<5 in a box, please specify Fisher's exact test p value or present only the results of the Fisher's exact test

-       Line 127: Table 2 title : specify n=161

-       Table 3 is redundant with Table 2 and should be removed.

Line 151 : Table 4 title : specify n=161

Table 4: Due to small numbers, present only the results of the Fisher's exact test

Line 153 : Table 5 title : specify n=161 and factors associated with malaria chemoprophylaxis use

Table 5: P values must be calculated and given for each variable, not for each modality of variables

Discussion:

Limits of the study were insufficiently discussed. As there is no control group, the study is not randomized or double-blind, and the numbers are small, without remote follow-up. Results must be modulated and the observational characteristics of this study taken into account in the discussion, particularly about neuropsychological side effects of mefloquine.

Results about length of stay were not discussed although this factor is usually described as a factor for non compliance among travelers. This point should be discussed.

Line 229: “However, after adjusting for these factors in multiple logistic regression analysis, the effect was not significant” This was not explained in the results or method section. Please consider to add it.

No conclusions is given. Please add one.

Author Response

I would like to express my sincere gratitude to the two reviewers for their professional analysis and constructive suggestions to make my paper more readable, and for the opportunity to revise. The reviewers’ suggestions and my revisions are outlined as follows. All emendations are marked in highlighted words in the text.  

Outline of Reviewers’ suggestions and the Author’s Correction

Reviewer #2

  • a better discussion of the limitations of the study and adding a conclusion.

The section was rewritten, with some topic sentences added and paragraphs revised for a clearer discussion of limitation. Conclusion was added.

  • Multiples citations should follow usual rules. For example [17] [18][19] or [30,31,32] should be cited as [17-19] or [30-32]

  Multiple citations are revised as recommended.

  • Abstract : Line 18-19 : « Of the 161 enrolled travelers, 103 (64%) reported side effects ». This appears to be in contradiction with the results (line 124) « Concerning the frequency of reported side effects, 66 of the 161 travelers (40.7%) reported at least one side effect ». Can you explain the difference or correct the sentence?

  103 of all the travelers didn’t report side effects. 58 of the 161 travelers (36%) reported at least one side effect. The mistaken numbers are corrected.

  • Line 107-108: « Overall, 32.7% of the travelers traveled to malarious regions, and 32.7% took antimalarial medications accordingly. » Could you please specify 32.7% of which population? Line 116-117 :

  The correct rate is 21.7% (35/161), the mistaken writing is corrected.

  • Table 1 title : specify n=161

  N=161 is specified at Table 1.

  • Table 1: Replace doxycycline by Doxycycline; female by Female.

  Replacement is done.

  • Side effects : - If so, specify that no severe side effects were observed or describe it.

  Description is added at table 2.

  • Define in methods how was calculated the score.

  Definition of the score is added below table 2.

  • In the table, the total amount of travelers with score>0 is 58 and you say in the text that 66 of the 161 travelers reported side effects : What can explain the difference ?

  The 66 is before exclusion of travelers on malarone and not completed questionnaires. In included travelers, 58 out of 161 travelers reported at least one side effect.

  • P values: when n<5 in a box, please specify Fisher's exact test p value or present only the results of the Fisher's exact test

  As suggested, all the results are presented by use of Fisher’s exact test.

  • - Line 127: Table 2 title : specify n=161

   Specification is done as advised.

  • - Table 3 is redundant with Table 2 and should be removed.

   Table 3 is removed for the above reason.

  • Line 151 : Table 4 title : specify n=161

  Specification is done as advised.

  • Table 4: Due to small numbers, present only the results of the Fisher's exact test

  All the results are presented with Fisher’s exact test.

  • Line 153 : Table 5 title : specify n=161 and factors associated with malaria chemoprophylaxis use

  Correction is done as suggested for better presentation.

  • Table 5: P values must be calculated and given for each variable, not for each modality of variables

  We adopt one of the variables as baseline to compare the odd ration and P values.

  • Discussion: Limits of the study were insufficiently discussed. As there is no control group, the study is not randomized or double-blind, and the numbers are small, without remote follow-up.

 Limitation of this study is added and discussed as the reviewer suggests.

  • Results must be modulated and the observational characteristics of this study taken into account in the discussion, particularly about neuropsychological side effects of mefloquine.

  The observation characteristics of this study is discussed, especially during explaining the neuropsychological side effects of mefloquine.

  • Results about length of stay were not discussed although this factor is usually described as a factor for non compliance among travelers. This point should be discussed.

  The influence on CP compliance of the length of stay is presented in the discussion part.

  • Line 229: “However, after adjusting for these factors in multiple logistic regression analysis, the effect was not significant” This was not explained in the results or method section. Please consider to add it.

   This sentence is deleted.

  • No conclusions is given. Please add one.
  •     Conclusion is added.

Round 2

Reviewer 1 Report

The authors have addressed all the concerns raised in the previous version of the manuscript and the quality has improved after incorporating required modifications. Therefore, the manuscript may be considered for publication in this Journal.